# A Bayesian semi-parametric model for thermal proteome profiling

Siqi Fang[1,4], Paul D. W. Kirk[2,3], Marcus Bantscheff [5], Kathryn S. Lilley [1,4✉] & Oliver M. Crook [1,2,4✉]

The thermal stability of proteins can be altered when they interact with small molecules, other biomolecules or are subject to post-translation modifications. Thus monitoring the thermal stability of proteins under various cellular perturbations can provide insights into protein function, as well as potentially determine drug targets and off-targets. Thermal proteome profiling is a highly multiplexed mass-spectrommetry method for monitoring the melting behaviour of thousands of proteins in a single experiment. In essence, thermal proteome profiling assumes that proteins denature upon heating and hence become insoluble. Thus, by tracking the relative solubility of proteins at sequentially increasing temperatures, one can report on the thermal stability of a protein. Standard thermodynamics predicts a sigmoidal relationship between temperature and relative solubility and this is the basis of current robust statistical procedures. However, current methods do not model deviations from this behaviour and they do not quantify uncertainty in the melting profiles. To overcome these challenges, we propose the application of Bayesian functional data analysis tools which allow complex temperature-solubility behaviours. Our methods have improved sensitivity over the state-of-the art, identify new drug-protein associations and have less restrictive assumptions than current approaches. Our methods allows for comprehensive analysis of proteins that deviate from the predicted sigmoid behaviour and we uncover potentially biphasic phenomena with a series of published datasets.

[1] Cambridge Centre for Proteomics, Department of Biochemistry, University of Cambridge, Cambridge, UK. [2] MRC Biostatistics Unit, School of Clinical Medicine, University of Cambridge, Cambridge, UK. [3] Cambridge Institute of Therapeutic Immunology & Infectious Disease (CITIID), Jeffrey Cheah Biomedical Centre, Cambridge Biomedical Campus, University of Cambridge, Cambridge, UK. [4] Milner Therapeutics Institute, Jeffrey Cheah Biomedical Centre, University of Cambridge, Cambridge, UK. [5] Cellzome GmbH, GlaxoSmithKline, Heidelberg, Germany. ✉email: ksl23@cam.ac.uk; omc25@cam.ac.uk

Thermal proteome profiling (TPP[1], also referred to as MS-CETSA) is a multiplexed mass-spectrometry extension of the cellular thermal shift assay (CETSA[2,3]). The guiding principle of these experiments is that heating generally causes proteins to denature and become insoluble. This heating can be performed at various temperatures and the remaining soluble protein quantified by mass-spectrometry (MS). This allows a temperature-solubility relationship to be determined and this is frequently called a melting curve[1]. The melting curve for each proteins is context specific and can be modulated upon binding to small molecules[4–6]. Thus by determining this melting curve for a large number of proteins in different contexts, for example in the presence of a drug, one can find targets and off targets of these molecules[1].

There are numerous applications of TPP and it is most commonly used to decipher drug-protein behaviours[1,5,7–12]. Moreover, it can be applied to study interactions with metabolites, nucleotides and nucleic acids[10,13–15]. Authors have shown that proteins in complex with each other are more likely to have concordant in vivo melting curves[16] and others have demonstrated that phosphorylation can alter thermal stability[17–19]. Thermal proteome profiling has also been complemented with extensive structural analysis[20–23]. Furthermore, TPP is not just applicable in human cells but can be applied in bacteria in vivo[12], in the apicomplexan parasite *Plasmodium falciparum*[14,24], and in tissue or blood[25]. Extensive work has recently been presented characterising the melting behaviour of proteins across 13 species, demonstrating similarities and difference for protein orthologues[26].

Thermodynamic theory predicts that the melting curve of proteins should have a sigmoid behaviour[27]. Melting curves of a protein may then be compared to determine context-specific behaviours. Statistical analysis can then follow a number of directions. For example, one approach involves summarising melting curves into a $T_m$-the temperature at which relative solubility has halved[1,5]. This is then followed by comparison of $T_m$ values across the two contexts using the appropriate z-score. This approach assumes that the melting curve is a bijection, else there might be multiple candidates for $T_m$. It also assumes that $T_m$ is defined, which need not be the case if relative solubility has never halved. Another approach is to compare the relative solubility at a fixed temperature[28]. However, summarising curves to a single value results in loss of information, loss of sensitivity and does not account for the quality of the fit of the parametric model[29]. A more powerful approach is to employ techniques from functional data analysis[30–32] and use the whole melting curve for statistics[29].

Childs et al.[29] introduced the method non-parametric analysis of response curves (NPARC) for powerful analysis of melting curves. In brief, the method assumes a sigmoid model for the data and then proceeds to perform an analysis of variance (ANOVA). Since typically TPP data involves measurement of melting curves for a great many proteins per experiment, the appropriate null distribution can be directly estimated from the data[33,34]. NPARC allowed thousands more proteins to be analysed than the original $T_m$ centric analysis and demonstrated a significant improvement in statistical power. However, this method still assumes a parametric sigmoid model and the method used to estimate the null distribution assumes that it is unimodal. Moreover, large-scale testing frameworks assume that the large majority of observations are samples from the null distribution, which can be problematic if the context of interest affects many proteins. Furthermore, there is no uncertainty quantification in the melting curves or the key model parameters.

To overcome these limitations, here we develop a Bayesian version of the sigmoid model, which allows uncertainty quantification. Furthermore, in the Bayesian framework one does not need to estimate the null distribution and multiplicity control is

automatic via the prior model probabilities[35–38]. In addition, including prior information on the model parameters has a number of benefits; allowing the shrinkage of residuals towards 0, the regularisation of the inferred parameters and improved algorithmic stability[39]. Through exploratory data analysis and model criticism, we find evidence for model expansion. We show that the standard sigmoid model is insufficient to model the relationship between temperature and relative solubility for some proteins. This motivates the development a semi-parametric model[40]. A semi-parametric model is one that includes both parametric terms, in our case the sigmoid, and unknown non-parametric terms. A Gaussian Process prior (GP prior[41]) is used to infer the non-parametric terms. Gaussian processes are highly flexible and have been used extensively in other molecular biology applications, such as gene-expression time courses[42–46], single-cell transcriptomics[47–49] and spatial proteomics[50,51].

Here we begin with exploratory data analysis of five datasets which motivates the creation of more flexible models. We then carefully analyse published data to demonstrate the improved sensitivity of our method, as well as the value of uncertainty quantification. Our proposed model can be applied more generally and we demonstrate, through simulations, that our approach has improved power and robustness to miss-specification of the parametric model. We identify putative protein–drug interactions that have been overlooked in previous TPP studies, including the protein HDAC 7 in studies designed to determine targets of the chemotherapeutic drug, Panobinostat. We proceed to characterise the proteins that deviate from sigmoid behaviour and uncover functional, as well as localisation, enrichments.

## Results

**Exploratory data analysis motivates model extension.** First, we interrogated data from five TPP experiments that were performed on the K562 human erythroleukemia cell line. The first experiment explored the effects of detergents on ATP-binding profiles. Then two other experiments explored the effects of different concentrations of the ABL inhibitor Dasatinib. In one of the experiments the histone deacetylase (HDAC) inhibitor Panobinostat was used to determine its effects on the behaviour of proteins. The final experiment explored the effects of the pan-kinase inhibitor Staurosporine. A summary of the experiments is given in Table 1.

We applied the NPARC pipeline to each of these experiments and carefully explored the results. The NPARC analysis approach makes a number of assumptions. Firstly, when estimating the null distribution, it assumes that the distribution is unimodal and thus a single F distribution is appropriate to approximate the null distribution. Secondly, it assumes that a large majority of the observed data are samples from the null distribution, which might not be the case for some contexts. For example, some highly indiscriminate ligands or perturbations that affected an entire organelle would violate these assumptions. Finally, it assumes that the sigmoid model is appropriate. To clarify, the 3-parameter sigmoid model of interest is the following:

$$S_{a,b,p}(T) = \frac{1 - p}{1 + \exp(b - \frac{a}{T})} + p. \tag{1}$$

The parameter $p$ is interpreted as a plateau, whilst $a$ and $b$ are shape parameters. This sigmoid model, and more generally sigmoid functions, makes the assumption of monotonicity, a single inflexion point, rotational symmetry around the inflexion point, a bell-shaped first derivative and horizontal asymptotes (at $p$ and $1 - p$). In many cases, such assumptions are appropriate and this behaviour is widespread in the TPP datasets we

**Table 1 Summary of the datasets and the respective reference used in this manuscript.**

| Dataset | Treatment | Concentration | number of proteins | Reference | Intact or Lysate |
|---|---|---|---|---|---|
| ATPdata | MgATP | 2 μM | 4177 | 8 | Lysate |
| Dasatinib 0.5 data | Dasatinib | 0.5 μM | 4625 | 1 | Intact |
| Dasatinib 5 data | Dasatinib | 5 μM | 4154 | 1 | Intact |
| Panobinostat data | Panobinostat | 1 μM | 3649 | 73 | Intact |
| Staurosporine data | Staurosporine | 20 μM | 4505 | 1 | Lysate |

examined (see Fig. 1C and E). However, we did observe proteins that deviated from this behaviour and violated these assumptions (between 3 and 20% depending on the dataset), beyond what could be attributed to measurement error. These include examples of a hyper-solubilisation phenomena; that is, proteins reproducibly increasing in relative solubility as temperature increases, which is not predicted by thermodynamics[27]. Maximum solubility would be expected at physiological pH and temperatures. We speculate that increase solubility with temperature might arise for various reasons. Firstly, some proteins may have insoluble sub-populations which are perturbed during the heating process. Indeed, we might be observing temperature-dependent phase transitions on a system-wide scale as noted previously by ref. [15]. Secondly, organeller membranes will be compromised in intact cells at higher temperatures resulting in some proteins undergoing conformational changes where the new conformation has higher thermal stability. Investigating these relationships further will require additional experimentation and is outside the scope of our study. Finally, technical issues such a variable co-isolation of TMT labelled peptides could also lead to an apparent increase in solubility of proteins with increased temperature, but we anticipate that this effect is minor.

After fitting a sigmoid model to each protein in each condition, we computed the residuals for every protein at each temperature. Classical analysis of variance assumes that the residuals are independently and normally distributed with homoscedasticity. We observed that none of these conditions are true for these data (see Fig. 1A for an example)[29] also noted this fact by comparing the empirically derived F distributions to those which would be obtained under classical assumptions and by also analysing the corresponding p-value histograms[52]. The significant departure of the F distributions from the theoretical behaviour was observed and so they used large scale data analysis tools to approximate the null. This results in different effective degrees of freedom for the F test and analysis of variance proceeds as usual. For sake of pedagogy, we state that bootstrapping or permutation methods, amongst others, could also have been used[34].

To perform residual analysis, we computed the sample Spearman correlation matrix for the residuals and observed that different datasets have different correlation structures (see Fig. 1B and C) and that residuals for closer temperatures are, in general, more correlated. The presence of correlated residuals usually suggests data structure that has not been correctly modelled[53–55].

To avoid estimating the null distribution, we recast the analysis of TPP data by proposing a Bayesian sigmoid model. This has the further benefit of allowing expert prior information to be included for the parameters. The Bayesian framework also allows us to quantifying the uncertainty in our parameter estimates and as a result the uncertainty in the fitted function. Given that we observed deviations from the sigmoid model and strongly correlated residuals, we proposed to include an additional functional term in our model. Given no suitable parametric candidate for this additional term, we sought inspiration from the Bayesian non-parametric literature and placed a Gaussian process prior on this additional term, allowing a more flexible set of

functions to be modelled and the uncertainty in this function to be quantified[56–58]. We refer to the methods section for a precise description of our model.

In the following sections, we focus more closely on the Staurosporine and Panobinostat datasets. The former is useful because Staurosporine is a pan-kinase inhibitor and we expect a large number of kinases amongst the true positive cases. As with previous authors, we use this as a pseudo-ground truth. For the other datasets true and false positive are poorly defined and we draw upon complementary literature in our discussions. We discuss all the datasets in collection in the final section and results are included as part of the supplement (see Supplementary data 1).

**Analysis of Staurosporine dataset.** Having developed sigmoid and semi-parametric Bayesian models, we applied these approaches to the Staurosporine dataset[1]. Staurosporine is a pan-kinase inhibitor, where the inhibition is achieved by a having high affinity to the ATP-binding site of kinases[59]. How Staurosporine affects the cell is not completely understood and has been shown to induce apoptosis[60] and cell cycle arrest[61]. The Staurosporine dataset that we consider reports relative solubility of proteins in the presence of 20 μM of Staurosporine for 2 control replicates and 2 treatment replicates. A total of 4505 proteins were measured using quantitative multiplexed TMT LC-MS/MS measurements at temperatures ranging from 37 degrees to 67 degrees in 10 even increments of 3 degrees[1].

One advantage of this dataset is that we expect a large number of kinases to be the target of Staurosporine. Hence, we might expect such proteins to have shifts in their thermal profiles upon Staurosporine treatment. Hence, as in previous analysis[29], we curate a set of proteins with the annotation 'protein kinase activity' from ensembl.db[62]. We then compute the sensitivity, the proportion of correctly identified positive cases, for the NPARC and two Bayesian, sigmoid and semi-parametric, approaches (taking the p-value threshold as 0.01 and, similarly, a posterior probability threshold as 0.99). The NPARC approach achieves a sensitivity of 33.7, whilst the Bayesian sigmoid model a sensitivity of 36.7 and the Bayesian semi-parametric model achieves 39.6 (see Fig. 2B). This suggests that avoiding estimation of the null and expanding the model flexibility can improve the sensitivity of the analysis. Unfortunately, in such cases specificity (the true negative rate), is not well defined, since proteins that are not kinases may also have their melting curve perturbed, perhaps due to changes in their phosphorylation state as a result of ablated kinase function[18]. We see similar improvements for sensitivity when considering other datasets (see Supplementary Note) and a simulation study is also included in the supplement demonstrating that the two Bayesian approaches outperform the NPARC method.

Improved sensitivity results in finding new proteins that are putative targets of Staurosporine. For example, DYRK1A, a dual-specificity kinase with both serine and tyrosine kinase activities[63,64], which is essential for brain development[65,66], was

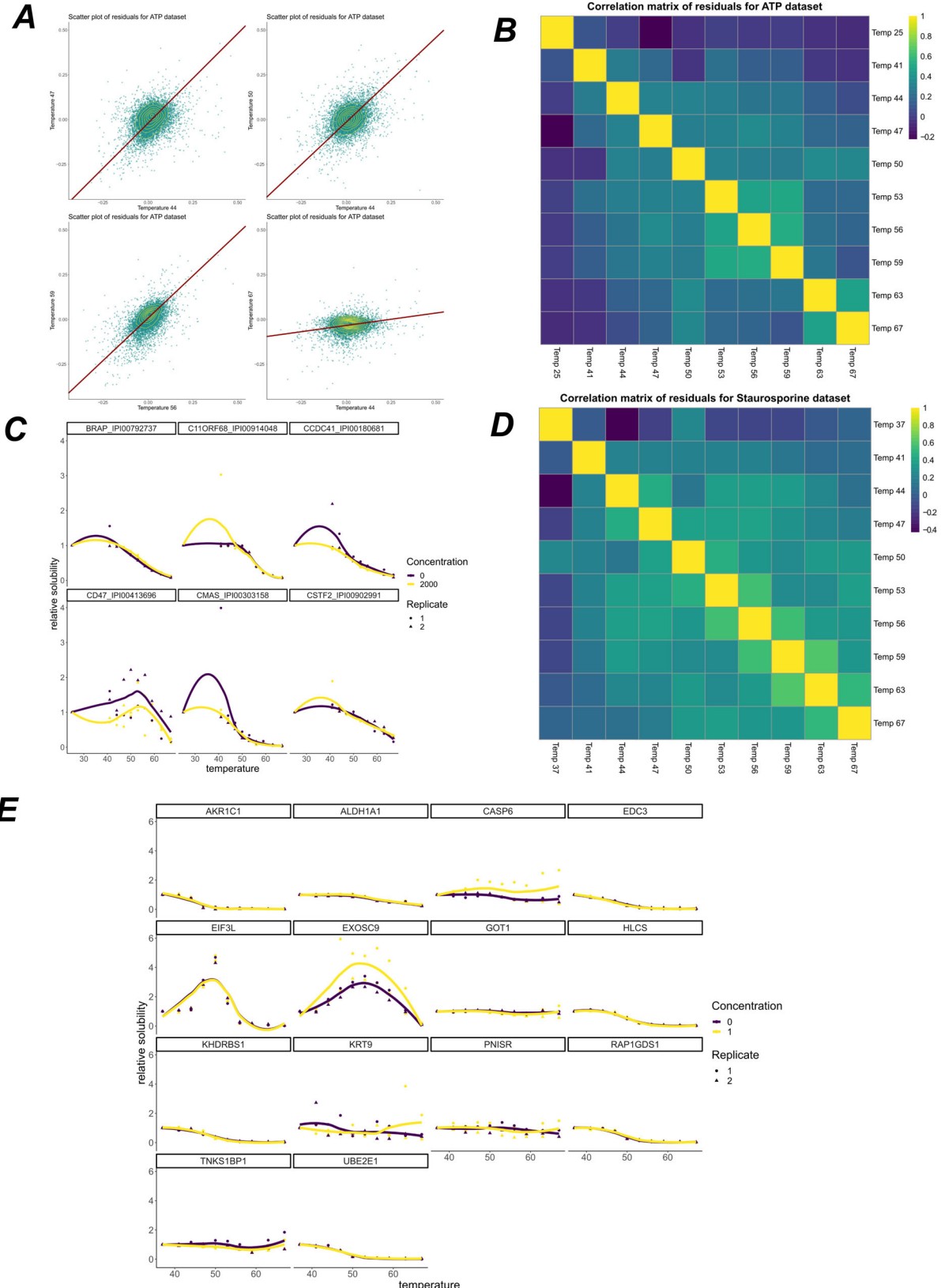

**Fig. 1 Residual analysis of thermal proteome profiling datasets. A** Scatter plots of residuals for the sigmoid model at different temperatures applied to the ATP dataset[8]. Orthogonal regression line shown in dark red and contours shown in yellow. Residuals are strongly correlated. **B** Sample Spearman correlation matrix of the residuals for the ATP dataset. **C** Example melting curves for some proteins from the ATP dataset. LOESS curves shown for visualisation. **D** as for (**B**), but for the Staurosporine dataset[1]. **E** Example melting curves from the Panobinostat dataset[73]. LOESS curves shown for visualisation. Concentration refers to Panobinostat concentration in μM.

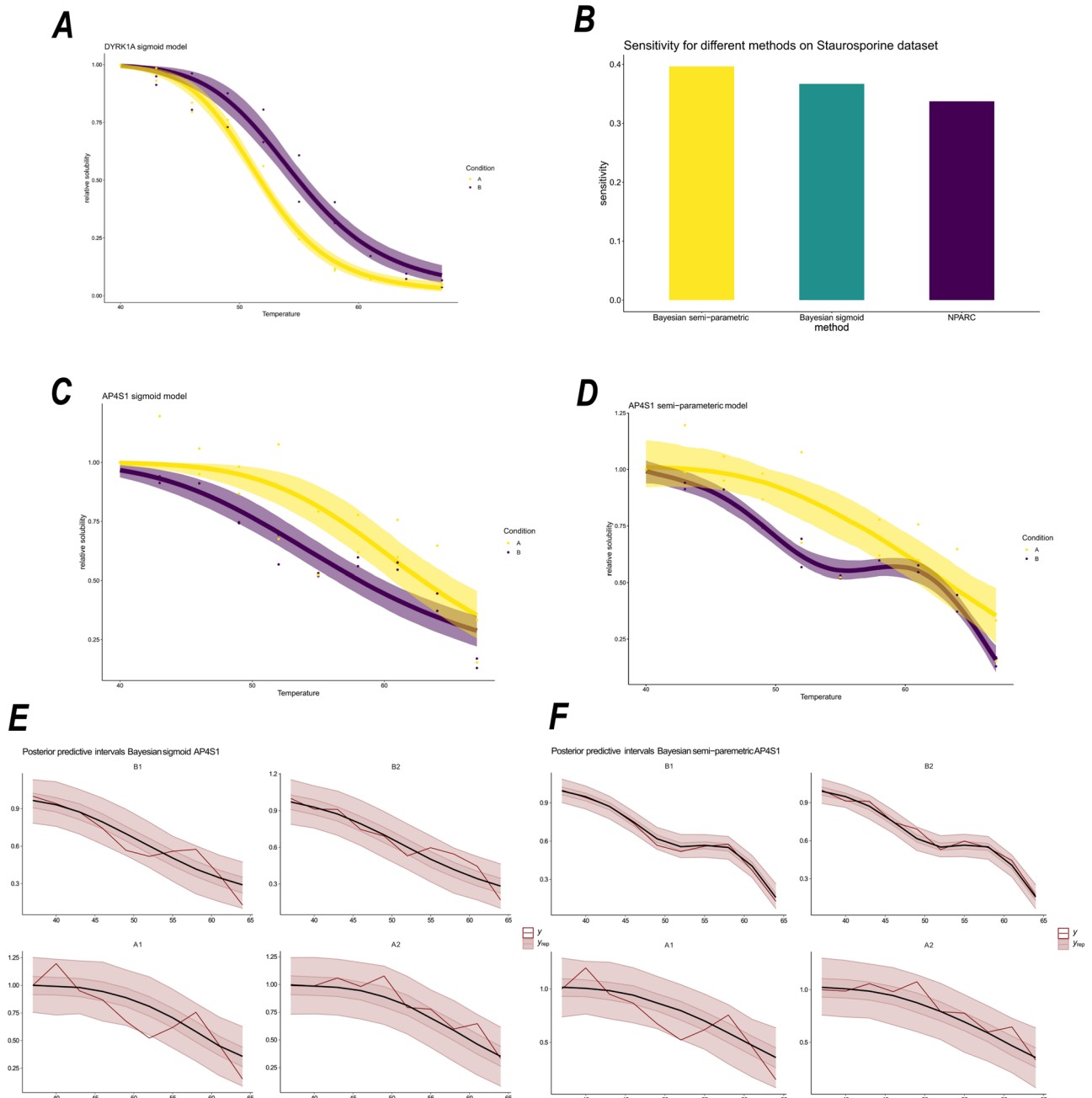

**Fig. 2 Analysis of Staurosporine dataset.** Condition A denotes the control and Condition B denotes 20 μM of Staurosporine (**A**) Melting profile for the DYRK1A with inferred mean sigmoid model function plotted, along with 95% credible bands for the inferred mean function. **B** Sensitivity for the different methods applied to the Staurosporine dataset. **C** Melting profile for AP4S1 using the sigmoid model, with uncertainty estimates in mean function. **D** Melting profile for AP4S1 using the semi-parametric model, including inferred mean function and 95% credible bands. **E, F** Posterior predictive checks for AP4S1 using the two Bayesian models: **E** sigmoid (**F**) semi-parametric. The red line corresponds to the observed data. Whilst the black line is the posterior predictive mean function and the credible bands correspond to 50% and 95% credible bands of the posterior predictive distribution, respectively. Statistics derived from two biological replicates, for each of two conditions each with 10 measure temperatures.

overlooked by the NPARC analysis. Our Bayesian analysis is able to determine DYRK1A as a kinase which is stabilised by Staurosporine (posterior probability >0.99). This observation is supported by kinobeads competition-biding experiments, where DYRK1A demonstrated a Staurosporine dependent effect (pIC$_{50}$ = 6.58)[67] and an isothermal shift assay (iTSA) also demonstrated a Staurosporine dependent effect on DYRK1A at 52 °C[28]. Figure 2A demonstrates other benefits of the Bayesian analysis, where we visualise uncertainty in the inferred sigmoid mean function. There is clear separation between the sigmoid curve

between the two conditions. However, it also highlights the potential limitations of the sigmoid model, with rotational symmetry imposed around the point of inflexion.

An even clearer example were the sigmoid model fails is the case of AP4S1, a component of the adaptor protein complex which is involved in vesicle trafficking from the trans-Golgi to the endosome[68,69]. Figure 2C shows the sigmoid model cannot model the multiple inflexion points of the melting curve of AP4S1. The limitation being the single inflexion point. Figure 2D shows the inferred mean function and associated uncertainty estimates.

Clearly the semi-parametric model is more appropriate for such cases. The full list of results can be found in the Supplementary material.

To compare these models more formally, we performed a posterior predictive check (see section 'Bayesian inference and model selection'). From the posterior predictive distributions, we examined the credible bands. To be precise, given a model, an observed value is predicted to fall in the credible band of size $\beta$ with probability $\beta$. Hence, if the observed data fall outside the credible bands, it is indicative of the model being insufficient. From Fig. 2E we see the data frequently lies outside the 50% credible band and occasionally outside the 95% credible band. Whilst for the semi-parametric model, visualised in Fig. 2F, the data never falls outside the 95% credible band and is more frequently contained in the 50% credible band. This suggests that the semi-parametric is more appropriate, in this case. Kernel density estimate based posterior predictive checks make a similar conclusion and are included in the supplement.

For a more quantitative treatment, we examine the out-of-sample predictive accuracy from the fitted Bayesian models (see section 'Bayesian inference and model selection'). We use leave-one-out cross validation (LOO-CV) with the log-predictive density as the utility function. Higher scores indicate better out-of-sample predictive performance. The LOO-CV estimate for the sigmoid model is $26.7 \pm 5.4$(SE), whilst for the semi-parametric model it is $41.1 \pm 6.5$ (SE). We conclude, for this protein (AP4S1), the semi-parametric model is superior. As a result of the improved modelling, our analysis was able to determine that AP4S1 was destabilised upon Staurosporine treatment (posterior probability >0.99), which we could not determine from NPARC or the Bayesian sigmoid model. AP4S1 is not a kinase, thus its change in behaviour upon Staurosporine treatment is not straightforward to interpret. In any case, we would expect kinases to be stabilised, rather than destabilised. This destabilisation might be an effect of not being correctly localised or not being able to correctly form a complex. AP4S1 localisation is dependent on the small G protein ARF1[70], whose function, it turn, depends on several kinases[71,72]. Thus, the destabilisation is likely a downstream effect of Staurosporine as a pan-kinase inhibitor.

**Proteins with altered thermal stability upon Panobinostat treatment.** The analysis of the Staurosporine dataset demonstrated the improved sensitivity of our method and the ability of our approaches to model complex behaviours, whilst also quantifying uncertainty. We next applied our method to the Panobinostat dataset where, in the original analysis, only a handful of hits were identified[73]. Panobinostat is a non-selective histone deacetylase inhibitor (pan-HDAC inhibitor) that is approved for use in patients with multiple myeloma[74]. Thermal proteome profiling was applied to K562 cells treated with a vehicle (control) or 1 µM of Panobinostat. 2 replicates in each context were produced and a total of 3649 proteins were measured[73]. These panobinostat experiments are cell-based rather than lysates and so we expect our approach to be sensitive to non-canonical melting curves that may be due to effects on solubility.

We applied the NPARC pipeline and identified 7 proteins as having their melting curve significantly altered ($p < 0.01$), which included the known Panobinostat targets HDAC 1, 6, 8, 10. The HDAC proteins are responsible for the deacetylation of lysine residues of the N-terminal of the core histones, as well as other proteins[75–79]. To quantify uncertainty, we applied the Bayesian sigmoid approach, also avoiding estimation of the null distribution. The Bayesian sigmoid model was able to identify 34 proteins whose melting profile was treatment dependent (posterior probability >0.99). 16 of these proteins are plotted in Fig. 3 and

these putative hits included all of the proteins discovered by the NPARC approach.

We also observed several proteins whose melting behaviour was not previously known to depend on Panobinostat; such as, NCBP1 whose behaviour appears to be destabilised upon Panobinostat treatment. NCBP1 is a nuclear cap-binding protein that is dual localised to the cytosol and nucleus, as well as being an integral component of the cap-binding complex[80,81]. Given the role of acetylation in formation of protein complexes[82], as well as NCBP1 having been shown to have two lysine residues that are substrates for acetylation[82] it possible that the observed melting behaviour is a downstream result of the ablated function of the HDAC proteins.

We have already demonstrated that non-sigmoidal behaviour is not unusual in the Panobinostat dataset (see Fig. 1E). Hence, we applied our Bayesian semi-parametric model to these data. We identified 85 proteins whose melting profile was panobinostat dependent with posterior probability greater than 0.99. These included HDAC 7, one of the core members of the histone deacetylation complex, which was not identified by either NPARC or the Bayesian sigmoid model (Fig. 4). In this case, however, HDAC 7 is not stabilised but, rather, destabilised suggesting indirect regulation downstream of Panobinostat targets. This finding is consistent with a recent report showing that HDAC7 abundance is regulated through activity of the known Panobinostat targets HDAC 1 and 3[83] and with HDAC 7 not being enriched in pull-down experiments with the Panobinostat[9].

Another protein that we identified with Panobinostat dependent behaviour was RUVBL1. RUVBL1 is a well-studied protein involved in histone acetylation and is a component of several complexes, has multiple localisations and many interaction partners[84–89]. RUVBL1 displays curious behaviour with both hypersolubilsation and destabilisation upon treatment with Panobinostat (Fig. 4). Since RUVBL1 has multiple states and is involved in multiple different complexes, it is possible that the effects of Panobinostat are interrupting only a certain pool of RUVBL1 proteins, leading to biphasic behaviour. Certain functional units of RUVBL1 might be more thermally stable than others, leading to complex temperature-solubility behaviours. The extent to which the behaviours are reflected in the melting curves will depend on many factors. Two dimensional thermally profiling experiments in lysate HepG2 cells show that RUVBL1 is highly thermal stable and did not display sigmoidal behaviour at several concentrations of Panobinostat (5, 1, 0.143, 0.02) µM at a temperature range of 42–63.9 °C[9].

**Characterising proteins that deviate from sigmoid behaviour.** Having established the utility of our Bayesian models, in particular the ability of our semi-parametric approach to model deviations from sigmoid behaviour. We next considered those proteins that were better modelled by the semi-parametric approach to see if they have any physical, functional or otherwise defining features. We began our investigation by selecting a set of proteins where the semi-parametric model explains at least 5% more variance[90] than the sigmoid model does alone (see Supplementary data 2).

We performed functional enrichment testing of these proteins using UniprotKB annotations (see supplementary data 3). We found that the post-translation modifications acetylation and phosphoprotein are enriched in these proteins across the 5 human datasets ($\forall i, _{pi} < 10^{-8}$ Fisher exact BH corrected), as well as RNA binding ($\forall i, _{pi} < 10^{-6}$ Fisher exact BH corrected). The pattern of enrichment can be visualised in Fig. 5A and is reproducible across all the datasets. Whilst the effect of

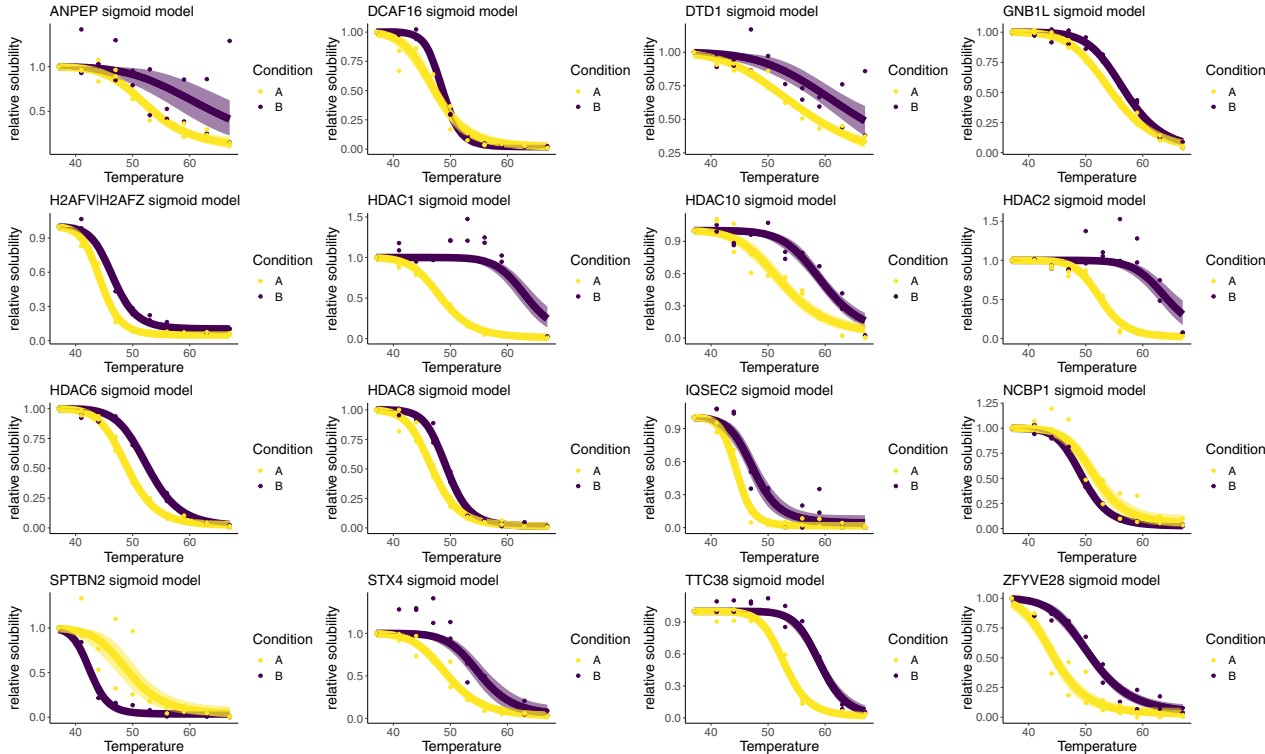

**Fig. 3 Example melting curves for Panobinostat dataset.** Melting profiles for 16 protein with posterior probability >0.99 in favour of a condition-dependent model using the Bayesian sigmoid model. Points are observed protein measurements. The inferred mean function from the sigmoid model is plotted as a line and the 95% credible band is given by the shaded region. Purple denotes the drug treated context, whilst yellow denotes the control. Statistics derived from two biological replicates, for each of two conditions each with 10 measure temperatures.

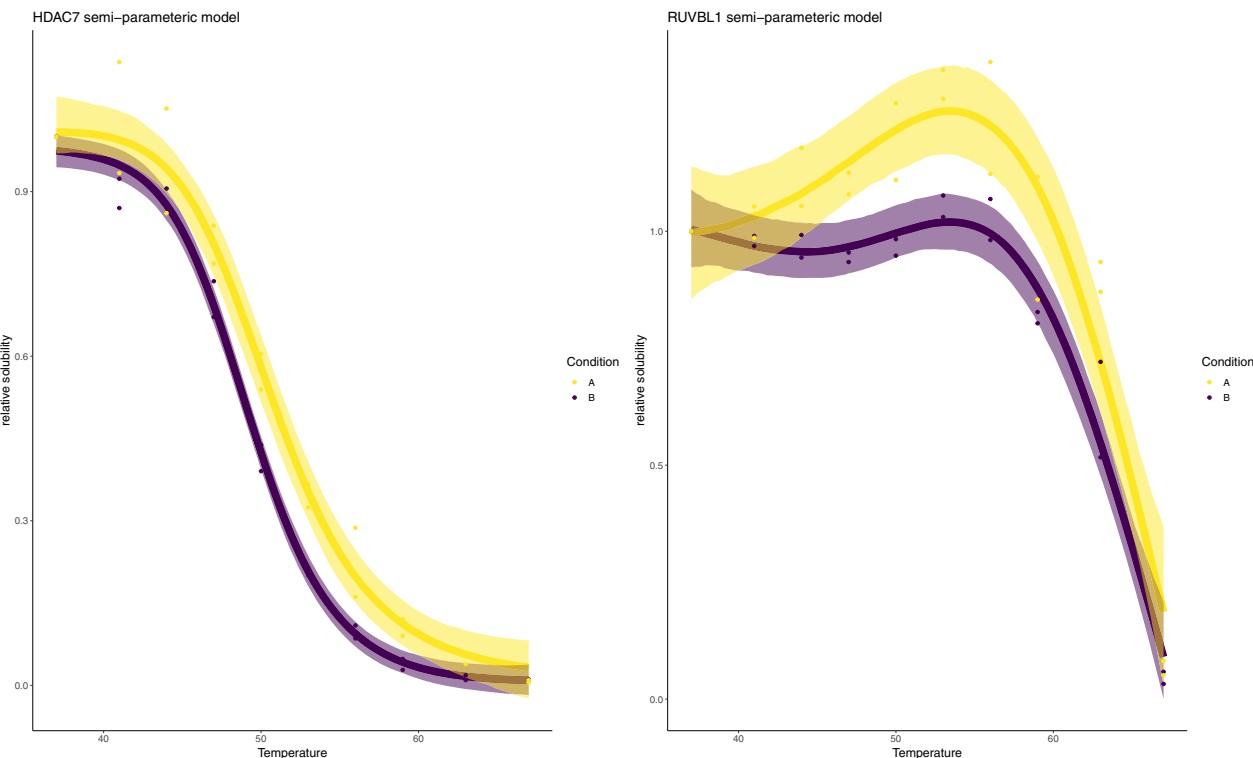

**Fig. 4 Example model fits using semi-parametric model.** Melting profiles for HDAC 7 and RUVBL1 using the Bayesian semi-parametric model. The points are observed protein data. The line represents the inferred mean function and the shaded region is the 95% credible band for the inferred mean function. Purple denotes the drug treated context, whilst yellow denotes the control. Statistics derived from two biological replicates, for each of two conditions each with 10 measure temperatures.

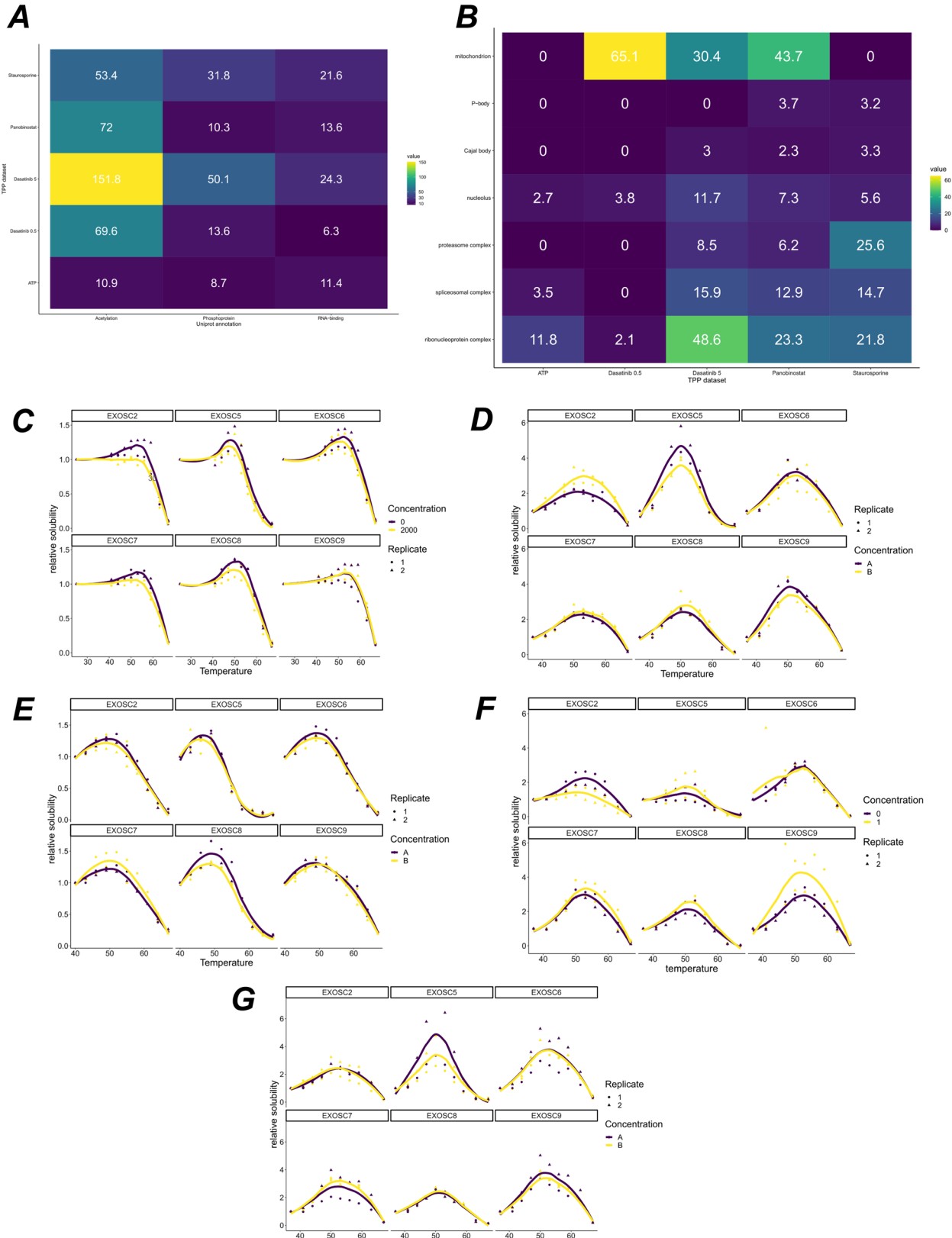

**Fig. 5 Enrichment analysis of protein deviating from sigmoid behaviour. A** Uniprot key term enrichment analysis. A tile plot show $-\log_{10}$ of the $p$-values for each of the terms for the 5 human datasets. **B** GO CC enrichment analysis. A tile plot showing $-\log_{10}$ of the $p$-values for each of the terms for the 5 human datasets. **C–G** Melting profiles of the proteins from the EXOSC complex, across the 5 human datasets, **C** ATP dataset (**D**) Dasatinib 5 dataset (**E**) Staurosporine dataset (**F**) Panobinostat dataset (**G**) Dasatinib 0.5 dataset. Statistics derived from two biological replicates, for each of two conditions each with 10 measure temperatures.

phosphorylation on protein thermal stability is well appreciated[18], the role of acetylation on thermal stability has not been characterised, despite well-established influence on protein stability[82]. Enrichment of acetylated proteins could suggest a mechanistic effect of acetylation on thermal stability.

Non-canonical melting behaviour may represent different pools of the same protein behaving differently within the cell. Non-canonical proteins are enriched for RNA-binding proteins and so the different species of protein, i.e. the RNA-bound form or the entities not bound to RNA, might have different temperature-solubility relationships, as well as different drug induced behaviours. Hence, what we may be observing in TPP datasets is a mixture of these behaviours being reflected in different ways. The extent to which one observes such behaviours will depend on the relative number of copies of each protein in each state and also on the particular way the modification effects the thermal stability of the protein. Hence, exactly which protein display this behaviour will be cell line and context specific, and so requires further investigation. This interpretation would explain both the hypersolubilisation and biphasic behaviour we have observed.

We continued to characterise the subcellular localisations of these proteins, with the hypothesis that these protein might come from a single or perhaps multiple localisations. As we see from Fig. 5B, the pattern for subcellular localisation is much less consistent than the pattern for functional enrichment and only the nucleolus and the ribonucleoprotein complex are enriched annotations for protein with non-sigmoidal behaviour in all the human datasets.

The nucleolus is a phase-separated sub-nuclear compartment and is the site of ribosome biogenesis[91]. Furthermore, during heat stress molecular chaperones accumulate in the nucleolus to protect unassembled ribosomal proteins against aggregation[92]. This effect is readily seen within 2 hours at 43 degrees. Despite TPP experiments usually only heating for minutes, we hypothesised that functional role of the nucleolus thus guards against the phenomena that TPP is attempting to induce. To test this hypothesis further, we filtered to proteins that are classed as non-sigmoidal and have known nucleolus annotation. We found that several proteins of the exosome complex EXOSC[2,5-9] fall into this class and are measured completely in all experiments. Figure 5 shows the reproducible non-sigmoidal behaviour. Remarkably, all members of this complex show hypersolublisation and increasing stabilisation until roughly 50 degrees. After 50 degrees the proteins destabilised. Without further experiments, we cannot deduce whether this effect is representative of the whole nucleolus or solely these EXOSC proteins. One alluring explanation could be that RNA dissociates from the EXOSC complex at 50 degrees. Furthermore, we do not observe significant co-aggregation of EXOSC protein in thermal proximity coaggregation (TPCA) data[16]. However, TPCA analysis derives curve similarity from an inverse euclidean distance, which may not be a sufficiently sensitive measure of curve similarity in this case.

Continuing our investigation into subcellular localisation, we integrated our analysis with spatial proteomics data from hyperLOPIT experiments[93]. We used hyperLOPIT data from U-2 OS cells, providing information on 4883 proteins to 11 subcellular compartments (refs. [94,95] and re-analysed in ref. [96] to reveal 14 compartments). We projected the proteins that deviate from sigmoid behaviour onto the PCA coordinates of the hyperLOPIT data (Fig. 6). In all datasets, we observed enrichment for nuclear, ribosomal and cytosolic regions, in agreement with our GO enrichment analysis. Furthermore, also in support of the GO enrichment results, we saw strong enrichment for mitochondrial annotations in the two Dasatinib datasets and the

Panobinostat dataset. To understand the functional relevance of these proteins, we stratified to the proteins that have mitochondrial annotations according to the hyperLOPIT data.

In the Dasatinib 0.5 dataset, we saw enrichment for cofactor binding ($p < 10^{-13}$), coenzyme binding ($p < 10^{-9}$), NAD binding domains ($p < 10^{-7}$), small-molecule binding ($p < 10^{-9}$), FAD binding domains ($p < 0.0001$), nucleotide binding ($p < 10^{-9}$), ATP-binding and RNA-binding ($p < 0.05$). We see similar results in the Dasatinib 5 dataset: cofactor binding ($p < 0.001$), coenzyme binding ($p < 0.001$), NAD binding ($p < 0.001$), nucleotide binding ($p < 0.001$), small molecular binding ($p < 0.01$). Almost identical results are seen for the Panobinostat dateset: cofactor binding ($p < 10^{-8}$), NAD binding ($p < 10^{-6}$), co-enzyme binding ($p < 10^{-6}$), small molecule binding ($p < 0.01$), nucleotide binding ($p < 0.01$), FAD binding domain ($p < 0.01$). Taken as a whole, these results support our interpretation of biphasic behaviour where different functional copies of a protein behave differently from each other and that we observe a mixture of these behaviours in TPP experiments.

Given the functional and localisation enrichments we have observed, we sought to further characterise these proteins by examining their intrinsic disorder. Indeed aggregation-prone proteins, after non-lethal heatshock, are enriched for intrinsically disordered regions[97]. Using the D2P2 database[98], we first obtained the length of the predicted intrinsic disorder regions (IDRs) for every protein. For stringency, we required that at least a minimum of four prediction tools were in agreement. To correct for length bias, we computed the proportion of the protein that was intrinsically disordered. We then tested if the set of proteins with non-canonical melting behaviour were enriched for proteins that had at least 5% of regions predicted to be intrinsically disordered. No such enrichment was observed (Fisher's exact test). We further filtered to proteins in our analysis that had nucleus annotations and despite nuclear annotated non-canonical proteins having a large proportion of IDRs (80–95%), there was no statistical enrichment beyond what one would have expected for nuclear proteins.

A further consideration is whether the experiment was performed in intact or lysed cells. Indeed, for the three experiments that were performed on intact cells (Dasatinib 0.5 and 5 and Panobinostat) the non-sigmoidal proteins showed an enrichment for mitochondrial localisation whilst the lysate-based experiments did not. In lysate-based experiments the mitochondrial membrane will break down and the local concentration of NAD will decrease. Hence, the drug has easier access to mitochondrial proteins in lysate-based experiments. Since cellular physiology is preserved for intact cells, we might believe that non-sigmoidal behaviour is indicative of downstream effects. However, some non-sigmoidal behaviours are reproducible and independent of whether the experiment was in lysed or intact cells. Thus, we cannot completely attribute these effects to whether the experiments were performed in intact cells or not.

## Discussion
We have presented Bayesian approaches to the analysis of thermal proteome profiling data. Our Bayesian sigmoid model quantifies uncertainty and avoids empirical estimation of the null distribution. The resulting model shows improved sensitivity and, as a result, we identified new putative targets and off-targets in 5 human TPP experiments. Uncertainty quantification provides useful additional information and, by inspecting the confidence bands, we can carefully select the temperatures at which to perform validation experiments.

Many proteins exhibit non-sigmoid behaviour and we observed strong correlation between residuals in all the datasets we

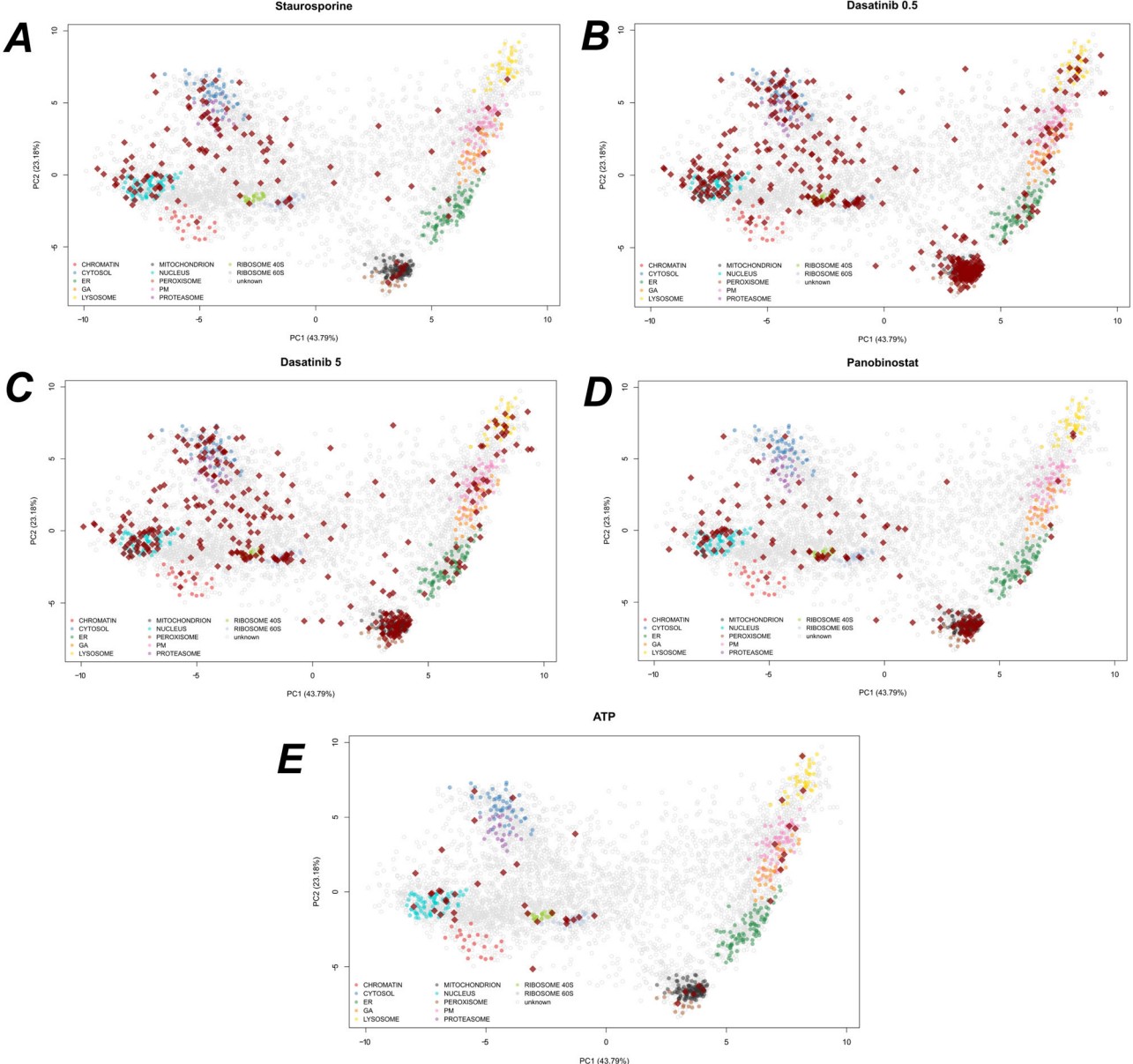

**Fig. 6 Subcellular localisations of proteins deviating from sigmoid behaviour. A–E** PCA plots of U-2 OS hyperLOPIT data[95], showing the top two principal components. Each pointer is a protein and marker proteins for each subcellular niche are coloured. Dark red diamonds denote proteins that were deemed to have non-sigmoid behaviour from TPP data. Each panel represents a different TPP dataset for the projected proteins. HyperLOPIT experiments are from biological triplicates with using a total of 57 fractions.

analysed, motivating an expanded model. Thus, we introduced a semi-parametric Bayesian model that further improved sensitivity, had better out-of-sample predictive properties for some proteins and had confidence bands with improved coverage. This improved analysis allowed us to identify HDAC 7 as having altered thermal stability on Panobinostat treatment, which previous analysis could not identify.

We probed the proteins that deviated from non-sigmoid behaviour and our analysis suggests that these proteins are enriched for proteins that contain known phosphorylation and acetylation sites, as well as RNA-binding proteins. These proteins also displayed concerted subcellular localisations with enrichments for nucleolus across all datasets and mitochondrion in particular contexts. This reinforces our interpretation that for proteins with non-sigmoid behaviour, we are observing a mixture of behaviours from different functional copies of those proteins.

This motivates expansion of the TPP method to deconvolute these behaviours, for example, phosphoTPP[17–19] and other PTMs. The RNA-binding behaviour could be examined with high-throughput RNA-protein enrichment methods[99] and further deconvolution could be obtained by combining TPP with spatial proteomics methods[93,95]. Though we observed non-sigmoidal behaviour in all datasets, more proteins were found to deviate in data generated from live cells (as compared to cell extracts).

As mentioned before, protein thermal stability can be affected by compound binding, PTMs and protein complex formation. In addition, protein solubility in cells might be affected by PTMs and other treatment-dependent effects, and even by ATP levels. Similar to protein solubility, compound treatment and other perturbations may affect the extent to which a protein is extracted in the applied experimental conditions leading to temperature

dependent and temperature independent components that manifest themselves in thermal denaturation profiles. Whilst most referenced studies have been directed at identifying direct targets of small molecule inhibitors in live cells or in cell extracts, there is an increasing recognition of the potential of TPP as a methodology to profile molecular phenotypes (e.g. ref. [100]) as it integrates multiple dimensions of regulation on proteome level into a single analytical approach. Such phenotyping could not only be informative for compound mechanism of action studies and to detect opportunities for combination treatments, but also to study effects of gene deletions, genetic variants and external stimuli and combinations thereof. As a consequence proteins can be affected in multiple ways and in different sub-cellular compartments resulting in more complex thermal denaturation behaviour than what can be robustly assessed with established computational approaches.

As demonstrated above our semi-parametric Bayesian approach is sensitive to detect protein effects that do not strictly follow the thermal denaturation-induced aggregation expected from isolated proteins and uniquely adds by identifying proteins affected by multiple parameters at once. Whilst not without challenges, the careful analysis of features in complex thermal denaturation curves is expected not only to facilitate hit calling but also to inform causality. This will be subject of future directions of our approach.

There are potential extensions of our methods to other TPP-based experimental designs[101], to simultaneous joint modelling of multiple organisms[26] and to include prior information derived from other experiments. We could also use expected gain in information to optimise the drug concentration and temperatures used in the TPP experiments[102]. Summarising and normalisation to protein-level could also be avoided by modelling the data at peptide spectrum match (PSM) level. We have also used a default global prior for the prior model probabilities - these might be better specified using known prior properties about the drug being used.

As with all methods, our approach is not without limitations, for example, increased computational cost could be a burden. However, if we are willing to sacrifice uncertainty quantification, we could simply use optimisation based inference instead. Our implementation is extensible with prior and model components easily change within our stan (probabilistic programming language[103]) implementation (see supplementary code).

## Methods

**Non-parametric analysis of response curves**. We briefly describe the NPARC method for completeness[29]. Let $y_{ijk}$ be the relative solubility of protein $i$ at temperature $T_j$ for replicate measurement $k$. The null hypothesis states that the relative solubility of protein $i$ at temperature $T_j$ is modelled as a single mean function regardless of the treatment condition or context:

$$\mathbb{E}(y_{ijk}) = \mu_i(t_j). \tag{2}$$

The alternative model allows for treatment effects or the mean function to change for each context

$$\mathbb{E}(y_{ijkc}) = \mu_{ic}(t_j) \tag{3}$$

where $c$ denotes the context. The mean function is modelled using the 3-parameter sigmoid model:

$$S_{a,b,p}(T) = \frac{1 - p}{1 + \exp(b - \frac{a}{T})} + p. \tag{4}$$

To clarify, under $H_0$ the parameters $a, b, p$ are fixed for both contexts, whilst under the alternative $H_1$ the parameters $a, b, p$ are allowed to be context specific. For hypothesis testing, the $F$ statistic is computed

$$F = \frac{d_2}{d_1} \frac{\mathrm{RSS}_0 - \mathrm{RSS}_1}{\mathrm{RSS}_1}, \tag{5}$$

where $\mathrm{RSS}_{0/1}$ denotes the sum of the squared residuals when fitting the null (0) or the alternative (1) model and $d_{1/2}$ are referred to as degrees of freedom. Large values of the $F$ statistic represents reproducible changes thermal stability. If the

residuals were *i.i.d* normally distribution then we could perform an $F$-test using the null distribution $F(d_1, d_2)$, where the degrees of freedom are computed from simple parameter and observation counting. However, the *i.i.d* assumption do not hold and so ref. [29] estimate the null distribution using new effective degrees of freedom $\tilde{d}_1, \tilde{d}_1$. Approximating the null distribution assumes a unimodal null distribution and that the majority of observations are samples from the null distribution. We refer to ref. [29] for detailed formulae. Once the approximate null has been obtained $p$-values can be computed as usual and multiple hypothesis testing correction applied[104].

### Bayesian inference and model selection

*Bayes' theorem and hypothesis testing*. In this section, we summarise Bayesian inference and model selection. The advantage of the Bayesian framework is that we no longer need to estimate a null distribution and multiplicity is automatically controlled via the prior model probabilities. This avoids making any assumptions about the properties of the null distribution. Furthermore, prior information is included on the parameters, which has a number of benefits, including allowing the shrinkage of residuals towards 0, regularising the inferred parameters and improving algorithmic stability. Furthermore, in a Bayesian analysis, we obtain samples from the posterior distribution of the parameters and hence the posterior distribution of the mean function can be obtained to quantify uncertainty.

Bayesian inference begins with a statistical model $\mathcal{M}$ of the observed data $y$ with the parameters of the model denoted by $\theta$. Given a prior distribution for the parameters, denoted $p(\theta|\mathcal{M})$, and observed data $y$, Bayes' theorem tells us we can update the prior distribution to obtain the posterior distribution using the following formula:

$$p(\theta|y) = \frac{p(y|\theta)p(\theta|\mathcal{M})}{p(y|\mathcal{M})}. \tag{6}$$

$p(y|\mathcal{M})$ is referred to as the marginal likelihood, since it is obtained by marginalising $\theta$:

$$p(y|\mathcal{M}_j) = \int_\theta p(y|\theta)p(\theta|\mathcal{M}) \, d\theta. \tag{7}$$

The task of hypothesis testing can be cast as a model selection problem. Indeed, the null hypothesis is associated with a model $\mathcal{M}_0$, whilst the alternative hypothesis is associated with a model $\mathcal{M}_1$. Thus, the task of hypothesis testing is that of selecting between two competing models.

To perform model selection, we are interested in the following posterior quantity[105],

$$p(\mathcal{M}_1|y) = \frac{p(y|\mathcal{M}_1)p(\mathcal{M}_1)}{p(y|\mathcal{M}_1)p(\mathcal{M}_1) + p(y|\mathcal{M}_0)p(\mathcal{M}_0)}, \tag{8}$$

that is the posterior model probability, given the data. The relative plausibility of two model is quantified through the posterior odds, which is the prior odds multiplied by the Bayes factor[106].

$$\frac{p(\mathcal{M}_1|y)}{p(\mathcal{M}_0|y)} = \frac{p(\mathcal{M}_1)}{p(\mathcal{M}_0)} \times \frac{p(y|\mathcal{M}_1)}{p(y|\mathcal{M}_0)} \tag{9}$$

The challenging of computing these equations is obtaining the marginal likelihood (equation (7)). We note that because of the integration with respect to the prior there is automatic penalisation of additional model complexity. The marginal likelihood is challenging to compute and is only available in analytic form for a small number of relatively simple models.

A number of sampling based approach are available to compute marginal likelihoods, such as bridge sampling[107,108], path sampling[109], importance sampling[110], harmonic mean sampling[111], nested sampling[112–114] (see also ref. [115]). Though these sampling based approaches produce highly accurate marginal likelihoods, these approaches require excessive computation in our case. Instead, we approximate the marginal likelihood using the Metropolis-Laplace estimator. Briefly, the log of the marginal likelihood (equation (7)) is estimated as ref. [116]:

$$\log(p(y|\mathcal{M}_j)) \approx \frac{P}{2}\log(2\pi) + \frac{1}{2}\log|\hat{H}| + \log(p(\hat{\theta}|\mathcal{M}_j)) + \log(p(y|\hat{\theta})), \tag{10}$$

where $\hat{\theta}$ a Monte-Carlo estimator of the parameters, $P$ is the number of parameters and $\hat{H}$ is estimated by the sample covariance of the posterior samples. This approach is used for both the Bayesian sigmoid model and the semi-parametric model.

Finally, we have yet to specify the prior model probabilities $p(M_j)$ for $j = 0, 1$. To control for multiplicity, we can adjust the prior model properties to assume that the null model is more likely that the alternative[35]. Hence, we set $p(\mathcal{M}_0) = 0.99$ and $p(\mathcal{M}_1) = 0.01$.

*Posterior predictive checks and out-of-sample predictive performance*. Formal model selection via the marginal likelihood can be used to select between two or more competing models. However, models can also be assessed and criticised using measures of predictive performance. Here, we consider posterior predictive checks, as well as out-of-sample predictive performance. A posterior predictive check

begins with simulating from the posterior predictive distribution:

$$p(\tilde{y}|y) = \int_\theta p(\tilde{y}|\theta, y)p(\theta|y) \ d\theta. \qquad (11)$$

This is the distribution obtain by marginalising the distribution of $\tilde{y}$ given $\theta$ over the posterior distribution of $\theta$ given $y$. The rationale is that simulated data from the posterior predictive should look similar to the observed data[39]. We simulate these datasets $y_{\text{rep}}$ and compute the 50% and 95% credible bands, for the models of interest. Though other posterior predictive summaries can be used, such as Kernel Density Estimate posterior predictive checks (see supplement).

Another approach is to examine the out-of-sample predictive accuracy from the fitted Bayesian models. We use (approximate) leave-one-out cross validation (LOO-CV) with the log predictive density as the utility function (equivalently the log-loss)[117]:

$$\text{ELPD}_{\text{LOO}} = \sum_{i=1}^n \log \int p(y_i|\theta)p(\theta|y_{-i}) \ d\theta. \qquad (12)$$

Equation (12) is the leave-one-out predictive density given the observed data without the *ith* observation, summed over the observations. This process is intensive so the expected log pointwise predictive density (ELPD) is estimated using Pareto smoothed importance sampling (PSIS)[117].

**Bayesian sigmoid model**. In this section, we develop our Bayesian sigmoid model. For our proposed Bayesian sigmoid model, we assume the aforementioned sigmoid model. As before, under $\mathcal{M}_0$ a single sigmoid model is posited irrespective of any treatment effects or contexts. While the competing model $\mathcal{M}_1$ allows the sigmoid parameters to be context specific. Thus under the null hypothesis, we assume

$$y_{ijk}|\mathcal{M}_0 \sim \mathcal{N}(S_{a,b,p}(T_j), \sigma_i^2) \qquad (13)$$

whilst for the competing model

$$y_{ijkc}|\mathcal{M}_1 \sim \mathcal{N}(S_{a_c,b_c,p_c}(T_j), \sigma_{ic}^2) \quad \text{for } c = 1, 2, \qquad (14)$$

where again $c$ denotes the context or treatment effect. To complete the specification of our model, we need to declare the priors. The sigmoid shape parameters $a, b$ are required to be positive and thus we place a Gamma distribution on these parameters. The right tail of the Gamma distribution discourages posterior mass on excessively large values of $a$ and $b$. To obtain reasonable defaults for these priors, we examined the fitted values found by previous analysis[29], as well as performing a prior predictive check[118]. Thus priors are specified for $a, b$ as follows

$$a \sim \mathcal{G}(7, 0.01) \qquad (15)$$

$$b \sim \mathcal{G}(7, 0.4). \qquad (16)$$

The parameter $p$ is restricted between 0 and 1 and thus a Beta prior is appropriate for this parameter. Given that the plateau is generally close to 0 and rarely above 0.5 we specify the following prior

$$p \sim \mathcal{B}(1, 20). \qquad (17)$$

For the standard deviation of the residuals $\sigma$, we desire these to be considerably smaller than the scale of the data and shrunk towards 0. This has two purposes: the first is that we want the data to be explained by variations in the mean function not simply by wide errors; secondly smaller residuals allow us to discriminate between small but reproducible shifts in melting profiles. We opt for the folded-normal distribution on $\sigma$[119]. We specify the prior as follows

$$\sigma \sim \mathcal{FN}(0, 0.05), \qquad (18)$$

which puts significant mass around 0 to encourage shrinkage, whilst residuals up to 0.4 are not considered surprising. There is no conjugacy between our prior and likelihood, which makes obtaining samples from the posterior distribution challenging. We employ Hamiltonian Monte-Carlo[120], in particular, a variant of the no-u-turn sampler[121,122] with an implementation in Stan[103,123].

**Bayesian semi-parametric model**. Our Bayesian sigmoid model allowed us to remove the assumptions relating to the estimating the null distribution, but still assumes a sigmoid functional form and uncorrelated residuals. To relax these assumptions, we propose a semi-parametric model. We assume the parametric sigmoid function and introduce an additional term so that the melting curves for protein $i$ are modelled according the following (suppressing notation on the condition)

$$y_{ik}(T_j) = S_{a,b,p}(T_j) + \mu_i(T_j) + \epsilon_{ij}, \qquad (19)$$

where $\mu$ is some deterministic function of temperature and $\epsilon_{ij} = N(0, \sigma_i^2)$ is a noise variable. Without any suitable parametric assumptions for $\mu_i$, we perform inference for $\mu_i$ by specifying a Gaussian process prior, so that:

$$\mu_i \sim GP(m(T), C(T, T')). \qquad (20)$$

A Gaussian process (GP) prior is uniquely determined by its mean and covariance function, which determine the mean vectors and covariance matrices of the associated multivariate Gaussians. We do not have any prior believe about the symmetry or

periodicity of our functions (beyond what is already encoded by $S_{a,b,p}$) and thus we specify a centred GP with a squared exponential covariance function

$$C = v^2 \exp\left(-\frac{\|T_i - T_j\|^2}{2l^2}\right), \qquad (21)$$

where $v^2$ is a marginal variance parameter and $l$, a length-scale parameter, encodes the distance at which observations are correlated. The adopted GP prior of $\mu_i$ tells us that the relative solubility for protein $i$ is modelled as follows

$$y_{ik}|S_{a,b,p}, \mu_i, \sigma_i \sim \mathcal{N}(S_{a,b,p} + \mu_i, \sigma_i^2 I_D), \qquad (22)$$

where $D$ denotes the number of measured temperatures. Note that we can make $n_i$ repeated measurement (or replicates) of protein $i$ at temperature $T_j$. We denote $y_i = \{y_{i1}, ..., y_{in_i}\}$ to be the concatenation of replicate measurements. Hence, the above implies that

$$y_i(T_1), ..., y_i(T_D)|\mu_i, S_{a,b,p}, \sigma_i \sim \mathcal{N}(f_i(T_1), ..., f_i(T_D), ..., f_i(T_1), ..., f_i(T_D), \sigma_i^2 I_{n_iD}), \qquad (23)$$

where $f_i(T_1), ..., f_i(T_D)$ is repeated $n_i$ times and $f_i(T_j) = S_{a,b,p}(T_j) + \mu_i(T_j)$. Our GP prior tell us that

$$\mu_i(T_1), ..., \mu_i(T_D), ..., \mu_i(T_1), ..., \mu_i(T_D)|v, l \sim \mathcal{N}(0, C_i), \qquad (24)$$

where $C_i$ is an $n_iD \times n_iD$ matrix. Note that the above means that we can marginalise $\mu_i$ to avoid inference of this unknown function and obtain:

$$y_i|S_{a,b,p}, v, l \sim \mathcal{N}(S_{a,b,p}, C_i + \sigma_i^2 I_{n_iD}). \qquad (25)$$

Reintroducing the context or treatment effect, we allow the parameters to vary between them. Thus, under the null hypothesis, we assume

$$y_{ijk}|\mathcal{M}_0 \sim \mathcal{N}(S_{a,b,p}(T_j) + \mu_i(T_j), \sigma_i^2) \qquad (26)$$

whilst for the competing model

$$y_{ijk}|\mathcal{M}_1 \sim \mathcal{N}(S_{a_c,b_c,p_c}(T_j) + \mu_{ic}(T_j), \sigma_{ic}^2) \quad \text{for } c = 1, 2. \qquad (27)$$

To complete our model, we need to specify the prior distributions. For parameters in common with the sigmoid model we make the same prior choices. Thus, it remains to make prior choices for $v$ and $l$. The challenges of specifying priors for the hyperparameters of the Gaussian process are well documented[124–128]. To obtain a sensible prior it is important to note that our model is weakly non-identifiable. This is because the non-parametric part can explain the parametric components. However, this is not, in general, an issue for Bayesian analysis. To advert problems this can cause for inference, we have to make judicious prior choices.

The first step is to encourage the marginal variance parameter to be on the scale of the residuals rather than that of the data. We already placed a folded-normal prior on the measurement error $\sigma$. For the marginal variance $v^2$, we impose even stronger shrinkage towards 0 by using a folded-student-t prior. This prior also has heavy tails allowing the non-parametric term to explain complex variations, if supported by the data. To summarise, we specify

$$v \sim \mathcal{FT}(3, 0, 0.5), \qquad (28)$$

where $\mathcal{FT}(\nu, m, \sigma)$ denotes a folded-student-t density with degrees of freedom $\nu$, mean $m$ and scale $\sigma$. On the other hand, for the length scale parameter $l$, we wish to avoid excessively small values. Short length-scales allow the Gaussian process simply to interpolate the data and overfit. Thus, we propose a log-normal prior for $l$, which has a sharp left tail and heavy right tail, discouraging small length scales and really large length scales, respectively. We find that the following prior works well in practice (sensitivity is tested in the supplement):

$$l \sim \mathcal{LN}(-0.5, 0.5). \qquad (29)$$

Inference for Bayesian models that incorporate Gaussian processes priors can be computationally intensive and so we make use of reduced-rank Gaussian process methods by approximating the covariance function[129]. As with the sigmoid model our semi-parametric model is implemented in Stan[103].

**Reporting summary**. Further information on research design is available in the Nature Research Reporting Summary linked to this article.

## Data availability
All data used in this manuscript are made available as part of the Supplementary material. Spatial proteomics data is available as part of the Bioconductor package pRolocdata. Python 2.7.15 was used to collect IDR data. String version 11.0 was used to collect enrichment data, which is available as Supplementary data 3. The remain data to reproduce the Figures is provided as Supplementary data 4.

## Code availability
The following version of R was used: r-3.6.1-gcc-5.4.0-zrytncq to analyse the data. Custom stan code was generated using version 2.21.2 and is provided as part of the Supplementary data 5.

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

## Acknowledgements

We thank members of the Cambridge Centre for Proteomics, Nils Kurzawa, and David-Paul Minde for insightful discussions. O.M.C. is a Wellcome Trust Mathematical

Genomics and Medicine student funded by the Cambridge School of Clinical Medicine. P.D.W.K. acknowledges MRC award MC_UU_00002/13. This work was supported by the National Institute for Health Research [Cambridge Biomedical Research Centre at the Cambridge University Hospitals NHS Foundation Trust] [*]. *The views expressed are those of the authors and not necessarily those of the NHS, the NIHR or the Department of Health and Social Care.

## Author contributions

S.F. and O.M.C. collected and analysed the data, developed the methods, and wrote the manuscript. O.M.C. and K.S.L. supervised the project. All authors interpreted the results and edited the manuscript.

## Competing interests

M.B. is an employee of GlaxoSmithKline. The remaining authors declare no competing interests.
