## [Peer Review File · Communications Biology]

Reviewers' comments:

Reviewer #1 (Remarks to the Author):

Brief summary:

Fang et al. present a study on mathematical modelling for thermal proteome profiling using a mass spectrometry-based approach, where it is assumed that any modification of the molecule might result in the alteration of the thermal stability, regardless of whether the protein undergoes PTM or through association with another component (protein, ligand, etc.). The data acquisition methodology cannot explain the alteration per se, unless it can be observed by mass shift of the measured peptides derived from the proteins and the associated adduct, such as the addition of a phosphate group, but in a case-control setting absolute quantification can be used to determine thermal stability by determining melting curves, and by comparison observe altered protein behaviour. Analysis of such data was originally performed based on thermodynamics which assumes sigmoidal temperature-dependent solubility, yet many exceptions to this were observed, and thus the authors propose an improved data analytics workflow using a Bayesian and semi-parametric Bayesian approach. They used five published datasets specifically collected for thermal stability studies and applied their methodology to this data.

Overall impression:

This manuscript describes an improved data analysis pipeline that certainly has merit and is of interest for mass spectrometry-based studies to uncover observable protein alterations upon system interference that are not apparent using traditional quantitative approaches. While it is arguable whether the datasets used are representative (as it appears that the Panobinostat datasets contains only two replicates and thus is statistically unsound) this should not detract from the improvements in mathematical modelling attempted in this study. I have gone through the maths and stats and to me it looks correct and mathematically sound. Yet it is not entirely clear how well the various methodologies truly perform other than the sensitivity and specificity values mentioned in the manuscript, and the authors should expand slightly more on this. They give examples where the semi-Bayesian approach is superior to the other two, but are there any instances where the semi-parametric Bayesian approach failed compared to the other ones? It is also noteworthy that, based on term-cluster analysis of the modelling output, the authors could infer sub-cellular locations or group functionalities where the system is perturbed, and thereby associate biological meaning to the data.

Specific comments:

General:

Text embedded in the Figures is not readily readable without excessive zooming. This should be changed to make it more accessible and readable.

Expand:

Page 4: "... , which might not be the case for some contexts." Please elaborate what these other contexts could be.

Page 14: "Non canonical": please include hyphen.

Page 15: "Figures 5 shows": change to "Figure 5 shows"

Page 15: "protein in (Thermal Proximity CoAggregation) TPCA data": please change to "protein in Thermal Proximity CoAggregation (TPCA) data"

Page 16: "non-sigmoidal behaviour is a indicative of downstream effects.": Change to "non-sigmoidal behaviour is indicative of downstream effects."

Page 19: "and combinations there-off.": change to "and combinations thereof."

Page 20: "(peptide spectrum match) PSM level.": change to "peptide spectrum match (PSM) level."

Page 20: "A Bioconductor package is in preparation": please include the tentative package and/or submitter name.

Unclear:

Page 4: "..., it assumes that the it is unimodal...": what do the authors mean with that?

Page 20: "within our stan implementation.": what do the authors mean with "stan"?

Figure 1E: what are the Concentration 0 and 1?

Figure2(A,C,D): Please expand on what Condition A and B mean.

Reviewer #2 (Remarks to the Author):

The paper describes Bayesian approaches to the analysis of Thermal Proteome Profiling (TPP), which allows uncertainty quantification and avoids empirical estimation of the null distribution. New model demonstrates better true positive rate on the data from 5 TPP experiments. The study addresses one of the weakest sides of conventional model that is disability to identify changes in proteins that show non-sigmoid behavior. The authors reported presence of correlated residuals using non-Bayesian approach that indicates need for better model. The sigmoid and semi-parametric Bayesian models were tested as more relevant for TPP. Although it is not very clear if the approaches cause critical increase of false positive results, this reviewer agrees the approach is worth to consider and publish in *Communications-Biology*. Thus, this reviewer suggests revising this manuscript prior to call final acceptance. Details are below

Comments:

1. Figures 1C, 5C. It is unclear why the melting curves start at nearly 25°C and no experimental points visible till 40°C.
2. Figure 1E, 5C-G. Authors should propose some suggestions why relative solubility of some proteins might be lower at physiological temperature than at > 40°C. It especially concerns intact cells experiment where maximal solubility is expected at physiological temperature range.
3. Chapter 2.3. TPP upon Panobinostat treatment. NPARC pipeline and identified 7 proteins with altered melting curves, Bayesian sigmoid model – 34, and Bayesian semi-parametric model – 85. There is no description of proteins that were identified using Bayesian approaches except 3 cases (NCBP1, HDAC7, RUVBL1). Authors should mention what are the rest of proteins, could they be false positives? Is there any sense to optimize Bayesian models in terms of false discovery rate decrease?
4. There is comprehensive description of analysis of Staurosporine and Panobinostat datasets by the Bayesian methods. Some description together with sensitivity measures should also be added for the rest 3 datasets mentioned in chapter 2.1.

Reviewer #1 (Remarks to the Author):

Brief summary:

Fang et al. present a study on mathematical modelling for thermal proteome profiling using a mass spectrometry-based approach, where it is assumed that any modification of the molecule might result in the alteration of the thermal stability, regardless of whether the protein undergoes PTM or through association with another component (protein, ligand, etc.). The data acquisition methodology cannot explain the alteration per se, unless it can be observed by mass shift of the measured peptides derived from the proteins and the associated adduct, such as the addition of a phosphate group, but in a case-control setting absolute quantification can be used to determine thermal stability by determining melting curves, and by comparison observe altered protein behaviour. Analysis of such data was originally performed based on thermodynamics which assumes sigmoidal temperature-dependent solubility, yet many exceptions to this were observed, and thus the authors propose an improved data analytics workflow using a Bayesian and semi-parametric Bayesian approach. They used five published datasets specifically collected for thermal stability studies and applied their methodology to this data.

We thank the reviewer for taking the time to read our manuscript carefully and provide useful feedback for improvement of our manuscript.

Overall impression:

This manuscript describes an improved data analysis pipeline that certainly has merit and is of interest for mass spectrometry-based studies to uncover observable protein alterations upon system interference that are not apparent using traditional quantitative approaches. While it is arguable whether the datasets used are representative (as it appears that the Panobinostat datasets contains only two replicates and thus is statistically unsound) this should not detract from the improvements in mathematical modelling attempted in this study. I have gone through the maths and stats and to me it looks correct and mathematically sound. Yet it is not entirely clear how well the various methodologies truly perform other than the sensitivity and specificity values mentioned in the manuscript, and the authors should expand slightly more on this. They give examples where the semi-Bayesian approach is superior to the other two, but are there any instances where the semi-parametric Bayesian approach failed compared to the other ones? It is also noteworthy that, based on term-cluster analysis of the modelling output, the authors could infer sub-cellular locations or group functionalities where the system is perturbed, and thereby associate biological meaning to the data.

Thank you for highlighting that the benefits other than sensitivity and specificity are unclear. Indeed, our semi-parametric Bayesian model requires less restrictive assumptions than the sigmoid-centric approaches. Furthermore, it can identify cases that deviate from the sigmoid behaviour and as the reviewer highlights these having meaningful biological annotations. We highlight these points in the text.

We would like to briefly clarify the two replicate issues. These data are functional, meaning that the observed data are samples from a curve (presumably sigmoid). 10 samples are made along

this curve in two independent biological replicates, totaling 20 observations. One could reduce the number of samples and increase the number of replicates but sufficiently sampling the curve is important because perturbation to the curve may be apparent at different temperatures. This has not thoroughly been explored in the literature in terms of how changing the design might affect power to reject the null hypothesis that the perturbation doesn't affect the melting profile. The case that has been explored in the literature is selecting one temperature, so-called isothermal profiling, and thus allowing more replicates but this also has limitations and has comparable power to the data in this paper. In any case, 2 replicates is sufficient to obtain biologically relevant hits, of course more replicates would render the analysis more powerful and allow more stable inference of variance parameters.

Through a number of simulations we tried to find cases where the semi-parametric Bayesian might fail. However, even in the case where the model was miss-specified (appendix 6), our model still performed well and we were able to control the false positive rate. One reason for this is that the semi-parametric model is a strict generalisation of the sigmoid model - the sigmoid model is a sub-model of the semi-parametric model. One challenge is to correctly specify the priors in the semi-parametric models but our careful strategy specified in the methods clearly works in a variety of situations. Another limitation is computational cost, the semi-parametric model is significantly more costly to fit but we have already highlighted this issue in the discussion.

Specific comments:

General:

Text embedded in the Figures is not readily readable without excessive zooming. This should be changed to make it more accessible and readable.

We apologize that the text in the figures was too small. We have resized for better clarity.

Expand:

Page 4: "... , which might not be the case for some contexts." Please elaborate what these other contexts could be.

We apologise for not being sufficiently clear. Null distribution approximation requires <5% of observations are truly perturbed i.e. a large majority of the observed statistics of interest are random draws from the null distribution. Some promiscuous ligands or perturbations may affect a large sub-proteome population- for example an entire organelle. These would violate that assumption.

Page 14: "Non canonical": please include hyphen.

Page 15: "Figures 5 shows": change to "Figure 5 shows"

Page 15: "protein in (Thermal Proximity CoAggregation) TPCA data": please change to "protein in Thermal Proximity CoAggregation (TPCA) data"

Page 16: “non-sigmoidal behaviour is a indicative of downstream effects.”: Change to “non-sigmoidal behaviour is indicative of downstream effects.”

Page 19: “and combinations there-off.”: change to “and combinations thereof.”

Page 20: “(peptide spectrum match) PSM level.”: change to “peptide spectrum match (PSM) level.”

Thank you for highlighting the above typos. They have been corrected.

Page 20: “A Bioconductor package is in preparation”: please include the tentative package and/or submitter name.

We have provided the methods as stan (probabilistic programming language) codes as part of the supplementary so others may use our method. Though an easy-to-use package is in development it is still preliminary, so we remove this statement to avoid misleading any readers.

Unclear:

Page 4: “..., it assumes that the it is unimodal...”: what do the authors mean with that?

Thank you for spotting the typo. The complete sentence should read: “Firstly, when estimating the null distribution, it assumes that the distribution is unimodal ...

Page 20: “within our stan implementation.”: what do the authors mean with “stan”?

Stan is the probabilistic programming language used to code our method and is provided as .stan files in the supplement. These can be compiled in R, python, Julia or from the command-line for easy use.

Figure 1E: what are the Concentration 0 and 1?

This is the concentration of panobinostat used in the experiment. We have clarified 0 or 1 μM in the legend.

Figure2(A,C,D): Please expand on what Condition A and B mean.

Condition A is the control and Condition B is in the presence of 20 μM of Staurosporine, we have clarified in the text.

Reviewer #2 (Remarks to the Author):

The paper describes Bayesian approaches to the analysis of Thermal Proteome Profiling (TPP), which allows uncertainty quantification and avoids empirical estimation of the null distribution. New model demonstrates better true positive rate on the data from 5 TPP experiments. The study addresses one of the weakest sides of conventional model that is disability to identify changes in proteins that show non-sigmoid behavior. The authors reported presence of correlated residuals using non-Bayesian approach that indicates need for better model. The sigmoid and semi-parametric Bayesian models were tested as more relevant for TPP. Although it is not very clear if the approaches cause critical increase of false positive results, this reviewer agrees the approach is worth to consider and publish in *Communications-Biology*. Thus, this reviewer suggests revising this manuscript prior to call final acceptance. Details are below

Thank you for the positive evaluation of our article and highlighting the value of our work in the current literature. We welcome the additional clarification to improve our manuscript.

Comments:

1. Figures 1C, 5C. It is unclear why the melting curves start at nearly 25°C and no experimental points visible till 40°C.

Thank you for highlighting this irregularity with the other datasets. We have provided the data TPP_ATP.csv file from the publication. It is unclear why the authors choose 25 degrees as their reference temperature. The second temperature appears to 41 degrees and no experimental value in between.

2. Figure 1E, 5C-G. Authors should propose some suggestions why relative solubility of some proteins might be lower at physiological temperature than at > 40°C. It especially concerns intact cells experiment where maximal solubility is expected at physiological temperature range.

The reviewers raise an interesting point. The reviewer is correct in suggesting that maximum solubility is expected at physiological pH. There are a number of possible reasons why protein solubility may increase with temperature:

Firstly, some proteins may have insoluble sub-populations under physiological conditions that show some increase in solubility during heating. Indeed, what our analysis may be revealing is temperature dependent phase transitions on a systems-wide scale as noted previously in Sridharan et al, doi: 10.1038/s41467-019-09107-y.

Secondly, at higher temperatures cellular substructures, such as organellar membranes will be compromised in intact cells, that could result in certain proteins undergoing conformational changes where the new confirmation has a higher thermal stability. This is partially borne out by

our observation from the non-sigmoidal behaviour of organelle proteins in some of the intact cell datasets, where changes in local concentrations of substrates may impact conformation and stability. Investigating these relationships further will require additional experimentation and is outside the scope of our study.

Moreover technical issues such as variable co-isolation of TMT labeled peptides could also lead to an apparent increase in solubility of proteins with increasing temperature, but we would anticipate that this effect would be minor and only significant in cases where few PSMs are assigned to a protein.

3. Chapter 2.3. TPP upon Panobinostat treatment. NPARC pipeline and identified 7 proteins with altered melting curves, Bayesian sigmoid model – 34, and Bayesian semi-parametric model – 85. There is no description of proteins that were identified using Bayesian approaches except 3 cases (NCBP1, HDAC7, RUVBL1). Authors should mention what are the rest of proteins, could they be false positives? Is there any sense to optimize Bayesian models in terms of false discovery rate decrease?

Thank you for drawing concerns about false positive identifications. The remaining proteins are provided as part of the supplementary material so that can be examined by others. The other proteins aren't mentioned in the text to remain brief. Indeed, some of these proteins will be false positives. There are a few separate ideas that are worth clarifying. Firstly, posterior probability is a slightly different concept to FDR, the former being a point property whereas the second is a set property. Indeed, say 5 proteins each have posterior probability 0.99 of being perturbed, then the probability they are all perturbed is $0.99^5 = 0.951$ - similar (but not equivalent) to a 5% FDR. Directly comparing FDR and posterior probability should be avoided to avoid conflating different concepts. The FDR rate should be considered as a global property of the entire set of proteins, whilst the posterior probability is a local property of each protein. For example the posterior probabilities would not change if we remove the top 20 proteins for the dataset, whilst the FDR certainly would. Though users should be confident in the ranking generated by our approach. The ranking generated by our posterior probabilities is clearly more statistically powerful as indicated through careful simulations in the supplement. Even in the case where the model is miss-specified, we have good control of the FDR and are significantly better than the NPARC model. Authors can easily select a probability threshold corresponding to a number of false positives they are willing to allow. One can also change the prior model probabilities if they wish to be more conservative. Finally, our method is sensitive to non-canonical changes in melting behaviour. These results have to be interpreted carefully because they are probably downstream effects rather than direct drug binding effects.

4. There is comprehensive description of analysis of Staurosporine and Panobinostat datasets by the Bayesian methods. Some description together with sensitivity measures should also be added for the rest 3 datasets mentioned in chapter 2.1.

Thank you for highlighting the lack of description for these other datasets. We focus on these datasets because there are proxies for true positives for the Staurosporine dataset (protein kinase activity) and for panobinostat there is a lot more complementary literature (such as pull-downs). The analysis of the ATP dataset was relegated to the appendix and for the Dasatinib dataset the true and false positive rates are poorly defined and so sensitivity measures are not appropriate. That is, without further experimentation, it is not clear what constitutes a true or false positive. Though these datasets are interesting from a modelling perspective and show concerted biology behaviour with other datasets. The Dasatinib experiments are also performed on intact cells which was important to study. The results are recorded in the supplementary data and we have added additional descriptions of these datasets to section 2.1.

REVIEWERS' COMMENTS:

Reviewer #1 (Remarks to the Author):

All points I raised were adequately addressed and mistakes corrected. I think the manuscript is now acceptable for publication.

Reviewer #2 (Remarks to the Author):

In this revised manuscript, the authors revised descriptions and address all concerns from this reviewer.